# Post Hoc Neuro-Symbolic Verification on Instruction Following of Language Models

## Abstract

Large Language Models (LLMs) are increasingly used for real-world problem-solving and decision-making. However, LLMs may not follow instructions, with subtle behavior that is hard to detect and diagnose. The impacts of instruction-unfollowing behavior may be further magnified in an LLM agent along its reasoning chain. This paper presents Nsvif, a novel framework for *post hoc* verification on instruction following of LLMs. At its core, Nsvif abstracts instruction-following verification as a Constraint Satisfaction Problem (CSP), where both instructions and LLM outputs are represented as structured constraints, including symbolic and neural constraints. Nsvif introduces a neuro-symbolic solver that embraces symbolic reasoning and neural inference—the former offers sound logic while the latter detects semantic violations. We curated a comprehensive benchmark, VifBench, to evaluate instruction-following verifiers, and developed a neuro-symbolic-guided synthesis method to construct data in a scalable and high-quality manner. We show the effectiveness of Nsvif on VifBench, where Nsvif significantly outperforms the existing baselines. Our work shows that unified symbolic verification with LLM-guided reasoning enables effective, reliable, and interpretable analysis of LLM instruction-following behavior.

## 1 Introduction

Large Language Models (LLMs) are increasingly used for problem-solving and decision-making in a wide variety of real-world tasks. Recently, LLM-based agents further organize individual LLM queries into complex, autonomous workflows. A core assumption behind these exciting use cases is that LLMs faithfully follow user instructions to make progress in the right direction. However, in practice, this assumption constantly fails—LLMs may misunderstand, partially follow, or even ignore critical parts of a given instruction (Jaroslawicz et al., 2025; Laban et al., 2025; Sirdeshmukh et al., 2025; Cemri et al., 2025). Such instruction unfollowing behavior could lead to unsafe decisions, incorrect outputs, and a loss of trust, undermining the safety and trustworthiness of AI technologies. In LLM agents, the impacts can be further magnified along the agent reasoning chain and workflow.

Despite significant efforts to align LLMs with human instructions through techniques like few-shot prompting (Brown et al., 2020; Lu et al., 2022; Agarwal et al., 2024), instruction tuning (Zhou et al., 2023a; Dong et al., 2023; Ding et al., 2023; Wang et al., 2023), and reinforcement learning from human feedback (Ouyang et al., 2022; Glaese et al., 2022; Bai et al., 2022; Cui et al., 2024), LLMs remain inherently probabilistic and lack *post hoc* guarantee of adherence. For example, GPT-5 can only correctly follow 69.6% of the instruction in the MultiChallenge benchmark (OpenAI, 2025; Sirdeshmukh et al., 2025); open-source models like GLM-4.5 and Qwen3-235B only follow up to 60% of the instructions (Team et al., 2025). Hence, effective *post hoc* verification is highly desired.

However, verifying whether an LLM follows user instructions is challenging. Natural language instructions are not always easy to check (e.g., "write a sentence in less than 200 words"), but can be ambiguous and open-ended ("write a creative, uplifting story"). In practice, we find that certain instructions can be mapped to formal semantics or symbolic constraints, while others require context-sensitive or semantic interpretation. A common practice in the field is to use LLM-as-a-judge (Zheng et al., 2023; Dubois et al., 2024; Li et al., 2024); however, such pure LLM-based neural approach often lacks accuracy and struggle to handle large, complex constraint spaces; meanwhile,

rule-based symbolic approaches are brittle and thus are limited in addressing the inherent ambiguity and under-specification in natural language instructions.

In this paper, we present NSVIF, a novel neuro-symbolic framework for *post hoc* verification on instruction following of LLMs. NSVIF formulates instruction-following verification as a Constraint Satisfaction Problem (CSP), where user instructions are encoded as structured constraint formulas and LLM outputs are encoded as variable values. This structured representation enables in-depth understanding of an LLM's reasoning, while preserving interpretability. NSVIF then introduces a neural-symbolic solver that embrace both *symbolic reasoning* and *neural inference* to solve the CSP. The former offers sound, logical reasoning for symbolic constraints that can be formally modeled, while the latter detects violations for neural constraints in the instructions. This hybrid design enables NSVIF to scale across diverse instruction types while maintaining accuracy and interpretability.

To comprehensively evaluate NSVIF as a verifier, we construct a new benchmark named VIFBENCH that covers different types of instruction-unfollowing behavior across multiple LLMs and application domains. Prior to our work, LLMBar (Zeng et al., 2024) is the only available benchmark for instruction-following verification. Unfortunately, LLMBar is too coarse-grained and lacks ground-truth labels on constraints. Each data point in VIFBENCH is created through a scalable pipeline that uses *symbolic synthesis* to generate abstract satisfiable logical formulas and *neural rewriting* to transform these formulas into natural-language problems and answers. The benchmark labels fine-grained constraints to support verification of both explicit and implicit instruction violations.

Our evaluation of NSVIF on VIFBENCH with state-of-the-art LLMs shows that NSVIF enables fine-grained constraint analysis and checking. For instructions with both formal and semantic constraints, NSVIF achieves $1.31\times$ pass@1 accuracy over LLM-as-a-judge approaches, while providing detailed constraint-violation information. In summary, this paper makes the following contributions:

- **New Principle.** Modeling instruction-following verification as a constraint satisfaction problem and solving it with a neuro-symbolic approach.
- **Framework and Tooling.** NSVIF, the first neuro-symbolic framework and toolchain that systematically checks LLM outputs against user instructions in natural languages.
- **New Dataset.** VIFBENCH, a novel benchmark and data synthesis toolchain for evaluating verification techniques of LLMs' instruction following.
- **Results.** NSVIF substantially outperforms baseline approaches, achieving higher precision in detecting instruction violations with interpretability.

## 2 BACKGROUND

Instruction following lies at the core of how users interact with LLMs. Given a natural language prompt or task description, users expect the model to generate outputs that are accurate, relevant, and aligned with the instruction. However, in practice, LLMs frequently exhibit instruction-unfollowing behaviors, where the generated output only partially satisfies—or completely deviates from—the user's intent (Jaroslawicz et al., 2025; Laban et al., 2025; Sirdeshmukh et al., 2025; Cemri et al., 2025). These violations may be explicit, such as generating the wrong function signature in code or producing output in the wrong format, or implicit, such as subtly misrepresenting facts in a summary or omitting constraints. Instruction-unfollowing is particularly problematic in software engineering and decision-making tasks, where correctness, determinism, and accuracy are essential. For instance, an instruction like "write a function that sorts integers in descending order" may be interpreted loosely by the model, resulting in ascending sort logic or unstable sorting behavior. In multi-turn agent settings, failure to follow a constraint in one step (e.g., not using a required API) can silently propagate, degrading the correctness of downstream actions.

Table 1 categorizes common types of instruction-unfollowing behavior of LLMs, distinguishing between those that can be symbolically verified through formal rules and constraints, and those that require neural methods to detect due to semantic or pragmatic complexity. The categories are derived from prior studies on instruction unfollowing behavior of LLMs (Zhou et al., 2023b; Chen et al., 2024; He et al., 2024; Jiang et al., 2023; Sirdeshmukh et al., 2025).

Figure 1 shows two examples of instruction unfollowing behavior. In Figure 1a, the LLM agent mistakenly deletes the data folder that the user instruction explicitly mentions not to delete, which

Table 1: Common instruction unfollowing behavior of LLMs.

| Dimension | Category | Description |
|---|---|---|
| Symbolic | Logical Constraint Violation | Output violates explicitly stated invariants, constraints, ordering, or uniqueness rules in the specification. |
| | Structural Violation | Output fails to adhere to the prescribed structural schema or data type (e.g., JSON, XML, domain-specific template). |
| | Invalid Element Error | Output omits required elements or includes prohibited entities explicitly stated in the specification. |
| Neural Network | Semantic Misinterpretation | Misunderstanding of the instruction's semantic content, ambiguity resolution failure, or subtle inconsistency in meaning. |
| | Pragmatic Mismatch | Output misaligns with the communicative intent, stylistic requirements, or permissible paraphrasing scope of the instruction. |

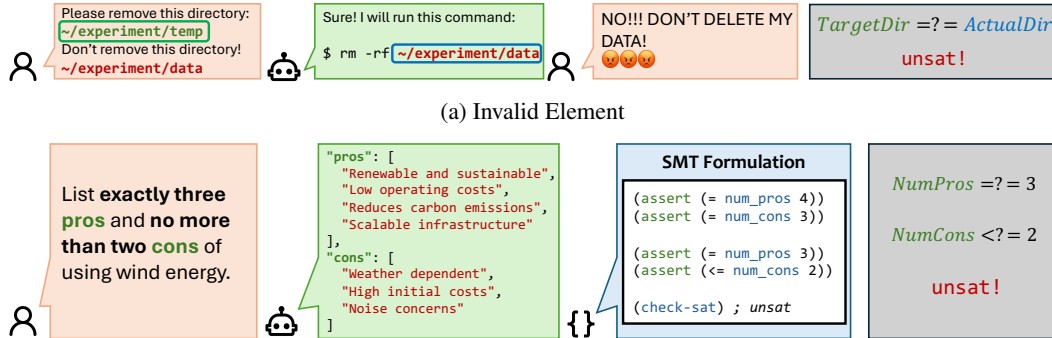

(a) Invalid Element

(b) Logical Constraint Violation.

Figure 1: Examples of instruction unfollowing behavior.

could lead to data loss. The pattern is referred to as Invalid Element in Table 1—despite that LLMs generate the right command, the target file path is incorrect. In Figure 1b, the LLM violates the logical constraints even though the instruction specifies them clearly—it generates too many pros and cons values. While such instruction unfollowing behavior may go unnoticed by human readers, especially in the context of autonomous agents, they are precisely the kind of errors that symbolic checkers such as SMT solvers can detect with certainty and minimal ambiguity.

## 3 NSVIF: NEURO-SYMBOLIC VERIFICATION OF INSTRUCTION FOLLOWING

The high-level idea of NSVIF is to synergistically combine the complementary advantages of symbolic logic and neural networks to verify whether the LLM's output follows a given user instruction. NSVIF builds on the observation that the requirements in user instructions vary widely in formalism and semantic clarity. Rather than forcing a one-size-fits-all solution, it categorizes user requirements into symbolic verifiable statements (e.g., code constraints, structured logic) and semantic directives (e.g., open-ended questions, stylistic preferences). Both of them can be modeled as *constraints*, with the former being symbolic constraints and the latter being neural constraints.

NSVIF uses a *dual-path neuro-symbolic verification framework*. It performs symbolic verification based on logical rules or constraint solvers to check whether the LLM's output satisfies the required conditions. For neural constraints where symbolic encoding is infeasible, it employs an LLM-as-a-judge to detect violations in the output. Figure 2 depicts an overview of NSVIF. NSVIF employs a three-phase approach to verifying if the LLM's output follows the instruction. The Planner analyzes constraints in a given instruction and generates individual verifier modules for the constraints. The Executor attempts to execute each verifier module, fixes any runtime errors, and gathers module results. Finally, the Solver formulates the instruction constraints into a Z3 program in Python. It then combines the constraints with module results to produce the final `sat`/`unsat` output.

This neuro-symbolic approach enables generalization across instruction types while preserving rigor and interpretability. The design also enables *modularity*, making NSVIF easy to extend—as

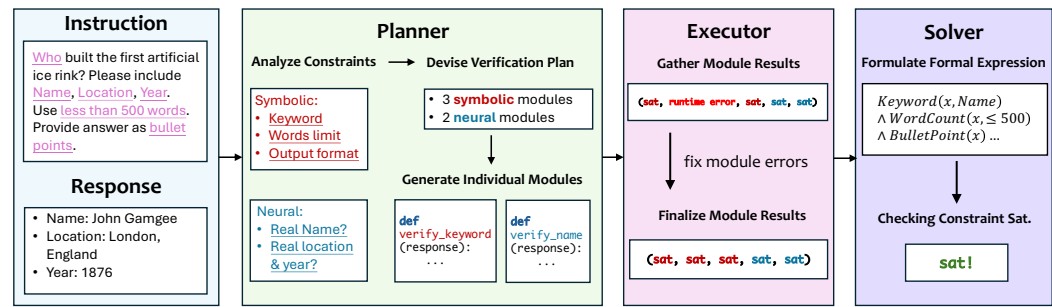

Figure 2: Overview of the NSVIF framework and the verification workflow.

more instruction types or domains emerge, the symbolic solver or neural verifier can be extended independently. With the decomposition of the verification problem, we aim to build a universal *post hoc* verifier that can be used to check instruction-following behavior.

## 3.1 FORMAL MODEL AS CONSTRAINT SATISFACTION PROBLEM

NSVIF models instruction following verification as a Constraint Satisfaction Problem (CSP). Let an instruction $I$ and model output $O$ define a verification instance $\langle I, O \rangle$. The goal is to determine whether $O$ satisfies the set of constraints implied by $I$, i.e.,

$$\text{verify}(I, O) = \begin{cases} \text{SAT} & \text{if } O \models \mathcal{C}(I) \\ \text{UNSAT} & \text{otherwise} \end{cases}$$

where $\mathcal{C}(I)$ denotes the set of constraints induced by instruction $I$. We define each instruction-induced constraint $\mathcal{C}_i(I), i \in 1, ..., n$ where $n$ is the number of induced constraints in the instruction, as:

$$\mathcal{C}_i(I) = \begin{cases} \mathcal{C}_i^{\text{sym}}(I) & \text{if } I \text{ can be formally specified} \\ \mathcal{C}_i^{\text{neu}}(I) & \text{otherwise} \end{cases}$$

That is, for each constraint, the verifier dynamically dispatches to one of two paths:

- **Symbolic constraints.** $\mathcal{C}_i^{\text{sym}}(I)$ can be explicitly encoded using logical rules, regular expressions, or executable specifications. Satisfaction is checked via symbolic reasoning or constraint solving.

- **Neural constraints.** $\mathcal{C}_i^{\text{neu}}(I)$ can be approximately checked by a neural verifier $\mathcal{V}_\theta(I, O)$, parameterized by prompt or model weights $\theta$:

$$\mathcal{V}_\theta(I, O) \in \{\text{SAT}, \text{UNSAT}\}$$

Note: although $\mathcal{C}_i^{\text{neu}}$ lacks explicit symbolic structure, it implicitly encodes a learned boundary between satisfying and violating outputs based on in-context examples or prompt-guided behavior.

This hybrid formulation allows NSVIF to treat verification uniformly as a CSP, and dynamically selects the symbolic or neural path for constraint evaluation based on the instruction type:

$$\text{verify}(I, O) = \begin{cases} \text{SAT} & \text{if } O \models \mathcal{C}_{\text{sym}}(I) \text{ or } \mathcal{V}_\theta(I, O) = \text{SAT} \\ \text{UNSAT} & \text{otherwise} \end{cases}$$

Based on the CSP formulation, we can systematically build both verifiers and benchmarks. For the verifier, the results of independently verified conditions outside the SMT solver, together with constraints that can be directly solved by a SMT solver, are encoded as a unified CSP problem solvable by the SMT solver. The solutions can be further post-processed and rewritten by an LLM to provide interpretable explanations. We describe such an implementation in §3.2.

To develop benchmarks, we can combine different symbolic constraint templates and neural condition templates according to predefined patterns. This process produces basic symbolic and neural instructions that are verifiable. The combined constraint patterns are then validated both by external verifiers and by the final SMT solver, with violations of individual conditions and combined constraints annotated accordingly. We developed such a benchmark in §4.

## 3.2 IMPLEMENTATION

NSVIF is implemented as a modular pipeline that converts natural language instructions and model outputs into verifiable CSPs. NSVIF is designed to be deployed in an LLM-powered workflow to inform the LLM whether it follows the user instruction, or as guardrails against LLM failures in agentic systems. Thus, three questions guide the design of NSVIF:

- **What** are the constraints in the instruction? **How** to check them?
- **Are** the constraints satisfied by **the output?**
- Combining each constraint's result, does the answer satisfy **all** constraints in the instruction? If not, **which** constraints are violated?

As shown in Figure 2, the implementation of NSVIF is composed by three components.

**Planner.** NSVIF uses a planner that guides the entire constraint verification process. The planner first translates both instructions and candidate outputs into structured logical forms.

- **Symbolic extraction:** For formally defined instructions (e.g., "Present in JSON format", "Write in less than 500 words", instructions are parsed into explicit predicates and constraints using a rule-based grammar augmented with a semantic parser.
- **Neural extraction:** For open-ended or ambiguous instructions (e.g., "Write a polite email"), we use a neural parser to analyze and parse the implicit constraints.
- **Constraint alignment:** Parsed instructions and outputs are normalized into a shared first-order logic representation made of logical predicates, ensuring compatibility with downstream solvers.

With the shared representation, for each constraint, the Planner generates an individual executable verifier module. Each module can individually verify one constraint. For formally defined, symbolic constraints, the Planner generates Z3 statements in Python that attempts to solve a system of formulas, where instruction constraints are expressed as symbolic formulas and the respective output restricts the variables' values. For subjective, fuzzy constraints, the Planner generates an LLM prompt that tailors to that constraints, which is then used to prompt an LLM for the verification result.

**Executor.** The modules generated by the Planner may encounter runtime errors when being executed. Thus, the Executor focuses on running, and if needed, fixing the generated modules. If the Executor observes that certain modules do not run, it enters a loop that attempts to fix the runtime error by prompting an LLM with the error message. Once all modules are finished, the Executor then collects the `sat/unsat` results and passes them to the Solver.

**Solver.** Finally, with all individual modules' results and the shared first-order logic representation of the instruction, the Solver attempts to verify whether the given output satisfies the instruction. Using the shared logical representation, the Solver generates Z3 statements in Python that represents the entire instruction and the provided LLM output. Each statement serves as a representation of the constraint. It signifies the constraint is either satisfied or not, based on the corresponding module's result. The Solver then adds all the statements into a Z3 `Solver()` object and runs the final Z3 program. The program produces both a binary decision (*followed / unfollowed*) and an *explanation trace*. Symbolic results highlight explicit violated constraints, while neural judgments include saliency-based rationales, enabling interpretability.

## 4 VIFBENCH: A BENCHMARK FOR INSTRUCTION-FOLLOWING VERIFIERS

Evaluating instruction-following verifiers like NSVIF would need a comprehensive benchmark. To our best knowledge, LLMBar (Zeng et al., 2024) is the only available benchmark that is related to the task. For instructions in LLMBar, we find that LLMBar does not provide ground truth of the output requirements (i.e., constraints), and thus cannot clearly assert whether an output of an LLM truly follows the instruction. Instead, LLMBar turns the verification problem into a classification problem. It presents two outputs to the verifier (or "LLM evaluator" in the LLMBar context) and asks it to choose which output better follows the instruction. There is no guarantee on whether or to what extent either of the two outputs follows the given instruction. Figure 3a shows the data schema and a data sample in LLMBar.

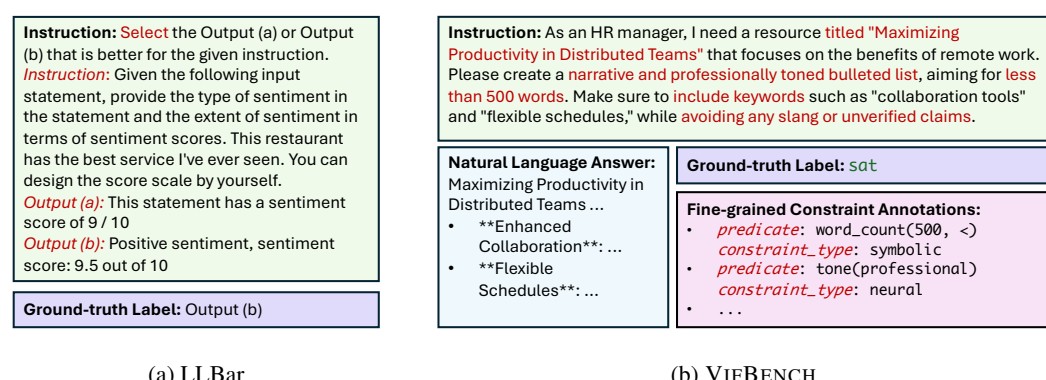

(a) LLBar        (b) VIFBENCH

Figure 3: Data schema and data sample of LLBar and VIFBENCH (our benchmark).

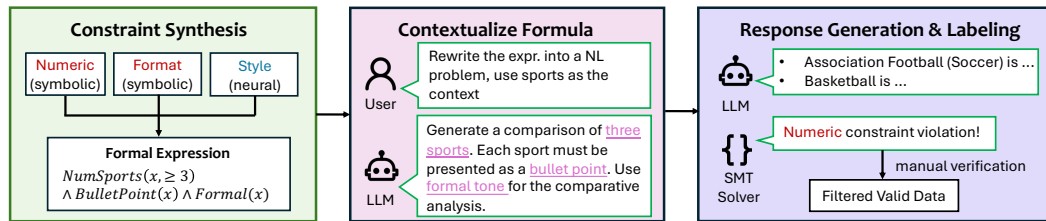

Figure 4: Overview of VIFBENCH's construction

To this end, we curate a new benchmark named VIFBENCH for instruction following verifiers. Instructions in VIFBENCH are synthesized based on rigorously specified constraints *a priori*. In this way, VIFBENCH can precisely evaluate an instruction-following verifier by checking whether it can comprehend the constraints in the given instruction and pinpoint the instruction unfollowing behavior. The data schema and a data sample of VIFBENCH is shown in Figure 3b.

Figure 4 shows the workflow of curating the dataset in VIFBENCH which consists of three phases.

**(1) Constraint Synthesis.** The lack of constraint annotations in an instruction hinders our ability to analyze an LLM evaluator's ability to correctly understand and analyze a given instruction. This makes it challenging to understand that when an evaluator incorrectly marks a result as instruction-following, whether the evaluator comprehends the internal constraints in an instruction. To fill this gap, each instruction in VIFBENCH originates from a shared internal abstract representation. We first collect generic logical predicates, such as $BulletPoint(x)$ or CNFs like $(a \lor b \lor \neg c)$. They serve as seed constraints for instruction generation. We say that a predicate is *symbolic* if it is completely logical, and a predicate is *neural* if it approximates real-life situations that require subjective judgment. These constraint predicates act as the logical foundations of instructions. It allows us to clearly define the expected answer criteria: any answer would need to satisfy the instruction's constraints. We then compose different constraint predicates together to form first-order logic formulas. We leverage existential and universal quantifiers in first-order logic to form complex dependencies between predicates. Each formula is checked by the authors to ensure that it is satisfiable. We also make sure the composed formulas do not contain unusual constraints that are hard to manifest in real-life LLM usages, such as $\texttt{FormalAudience}(text) \land \texttt{InformalTone}(text)$

**(2) Formula Contextualization** The abstract first-logic formulas clearly define the answer criteria, but it is too abstract for an LLM to generate an example answer. For each formula, we leverage a neural paraphraser to rewrite the formula into a natural-language instruction. The prompt we used is provided in Appendix B. For formulas composed of symbolic constraints, we prompt the neural paraphraser to rewrite it in a real-life context, such as meal preparation or travel planning. For formulas composed of neural constraints, the neural paraphraser is asked to rewrite under a similar context that fits the predicates themselves. Take Figure 4 as an example, the formula already presents a context, sports. The neural paraphraser then produces a natural-language instruction that conforms

to the formula which also allows an LLMto produce a valid answer, rather than providing the formula itself.

**(3) Response Generation and Labeling.** With the contextualized instruction, we prompt an LLM to produce an answer to the instruction. The LLM makes a faithful attempt at answering the instruction, but due to the LLM's inherent randomness, it cannot guarantee a correct answer, giving the benchmark answers that either `satisfies` or `unsatisfies` the instruction. Thus, for each instruction-output pair, we carefully analyze whether the output follows the instruction. For symbolic constraints, we leverage SMT solver to provide guarantees on satisfiability. For neural constraints, the authors decide on the satisfiability. Decisions are cross-checked and discussed between authors.

In summary, each natural-language instruction in VIFBENCH is clearly labeled with constraints on potential answers, such that if the verifier under test makes an incorrect decision, the user understands which constraint(s) the verifier has failed to understand and analyze. Due to the nature of the underlying logical formulas, even if an instruction can have multiple satisfying answers, they will be semantically the same.

## 5 EVALUATION SETUP

### 5.1 BASELINE

We use LLM-as-a-judge as a baseline of developing instruction-following verifier. LLM-as-a-judge is a common practice to classify unlabeled data based on fuzzy or underspecified criteria (Zheng et al., 2023; Zeng et al., 2024; Sirdeshmukh et al., 2025; Qin et al., 2024). We use a standard implementation of LLM-as-a-judge, where the LLM is given the instruction and the answer in VIFBENCH. The prompt then asks the LLM to decide whether the answer satisfies the instruction. The raw prompt can be found in Appendix A.

We run both NSVIF and LLM-as-a-judge on every instruction in VIFBENCH with GPT-4o (snapshot: gpt-4o-2024-08-06) and GPT-4o-mini (snapshot: gpt-4o-mini-2024-07-18). Additionally, we evaluated DeepSeek-V3.1 and DeepSeek-R1 on LLM-as-a-judge. While we intended to evaluate these two models on NSVIF, we could not complete the evaluation due to several intrinsic shortcomings of the models: (1) *infinite thinking:* As a reasoning model, DeepSeek-R1 occasionally becomes trapped in an infinite thought loop on particular tasks, thus failing to generate an output in the desired format. (2) *high-quality code generation capability:* For the NSVIF pipeline to run successfully, the **Planner** is required to generate correct code, or the **Executor** must fix incorrect code. This imposes a significant requirement on the model's code generation abilities. We found this to be a consistent issue with DeepSeek-V3.1, which frequently failed to produce executable code for certain data points.

Therefore, the final evaluation results for NSVIF do not include these models. We believe that as model capabilities improve, these issues will gradually diminish.

### 5.2 METRICS

We follow the pass@k evaluation proposed by Chen et al. (2021) and report pass@1 result:

$$\text{pass@}1 = \frac{1}{k}\sum_{i=1}^{k} p_i,$$

where $p_i$ denotes the correctness of NSVIF's verification on the $i$th data point against the ground-truth and $k$ denotes the number of data points in VIFBENCH. This metric serves as the most direct indicator of NSVIF's performance.

Moreover, we adopt the standard metrics of *Precision*, *Recall* and *F1 Score* to assess the effectiveness of NSVIF, which are defined as follows (Olson & Delen, 2008; Sasaki, 2007):

$$\text{Precision} = \frac{tp}{tp + fp} \times 100\%, \ \text{Recall} = \frac{tp}{tp + fn} \times 100\%, \ \text{F1 Score} = \frac{2tp}{2tp + fp + fn} \times 100\%$$

where $tp$, $fp$, $tn$, and $fn$ denote the numbers of true positives, false positives, true negatives, and false negatives, respectively. Consequently, the numbers of actual positives and actual negatives are equal to $tp + fn$ and $fp + tn$, respectively.

## 6 RESULTS

We evaluated NSVIF on in VIFBENCH with Pass@1, F1-Score, Precision, and Recall. The instructions in VIFBENCH contain both symbolic and neural constraints, approximating real-life LLM usages.

**How effective is NSVIF on verifying instruction following?** Table 2 shows NSVIF's and LLM-as-a-judge's results, where the best result is marked in **bold** and the second best marked in underline. In terms of general accuracy in verifying instruction following, NSVIF achieves at least $1.31\times$ higher Pass@1 accuracy compared with LLM-as-a-judge baselines. Similarly, NSVIF achieves at least $1.44\times$ higher *Precision* compared with the baseline verifiers.

Table 2: Performance of NSVIF and baseline on VIFBENCH.

| Method | Model | Pass@1 | Precision | Recall | F1 Score |
|---|---|---|---|---|---|
| LLM-as-a-judge | GPT-4o | 53.3% | 53.3% | **100%** | **69.6%** |
| | GPT-4o-mini | 53.3% | 53.3% | **100%** | **69.6%** |
| | DeepSeek-V3.1 | 53.3% | 53.3% | **100%** | **69.6%** |
| | DeepSeek-R1 | 53.3% | 53.3% | **100%** | **69.6%** |
| NSVIF | GPT-4o | **70%** | **76.9%** | 62.5% | 69.0% |
| | GPT-4o-mini | 40.0% | 43.8% | 43.8% | 43.8% |

NSVIF has a lower *Recall* and *F1 Score* compared with LLM-as-a-judge approaches. All LLM-as-a-judge achieves the same results. By analyzing responses from the LLMs, we found out that for all LLM-as-a-judges, they responded `sat` to *all the evaluated instructions*. This signifies that LLM-based verifiers do not understand instruction constraints at all, producing many false positives. One potential explanation for the uniform `sat` response is that the answer in the instruction-answer pair are structurally similar to a `sat` answer but itself is `unsat`. For example, certain instructions contain constraints on word counts (e.g., "less than 150 words"). Structurally, a writing response with 151 words looks the same as one with 149 words. This level of subtlety poses a great challenge for pure LLM-as-a-judge solutions to verify, as LLMs' inherent non-determinism prevents them from consistently reasoning about the instruction.

Answering `sat` to all instruction shows the danger of deploying LLM-as-a-judge approaches as safeguards in agentic systems. For safety purposes, a verifier with higher false negatives (i.e., the LLM `sat`isfies the user instruction but falsely marked `unsat`isfied) is preferred over one with higher false positives. In contrast, NSVIF's higher *Precision* and lower *Recall* demonstrate a tool towards applying instruction following verification on agentic systems. Although NSVIF reports more false negatives than the baselines, false negative inflicts less damage compared with false positives: in the same LLM workflow, a falsely marked `unsat`isfying response can simply be retried.

On the smaller model (GPT-4o-mini), NSVIF performs worse than LLM-as-a-judge. The reason is that the constraint analysis and code generation are challenging for GPT-4o-mini. NSVIF's Planner requires advanced logical reasoning ability to correctly analyze the constraints in a given instruction. It also requires the model to generate correct code for individual verifier modules. A less-advanced model cannot accomplish these tasks at the same time. Our future work includes exploring a hybrid approach where simpler tasks are offloaded to a smaller model, such as fixing runtime errors.

**How does NSVIF's performance vary with the number of constraints?** Figure 5 shows how NSVIF's and LLM-as-a-judge's performance varies by the number of constraints in an instruction. NSVIF and LLM-as-a-judge perform similarly with small-to-medium number of constraints. When the number of constraints reaches 9, NSVIF's three-phase approach allows more instructions to be correctly verified while LLM judges struggle. When the number of constraints increases, NSVIF's divide-and-conquer strategy allows NSVIF to verify each constraint individually, maintaining the LLM's focus on a single constraint. In contrast, LLM-as-a-judge's performance drops. This shows that while LLM-as-a-judge performs on par with NSVIF when instructions are simple, pure LLM-based approaches in such verification tasks cannot scale to complex scenarios. In contexts such as multi-agent systems, constraints can overlap and combine, forming complex dependencies. NSVIF's result shows the effectiveness of the CSP formalization under complex instructions.

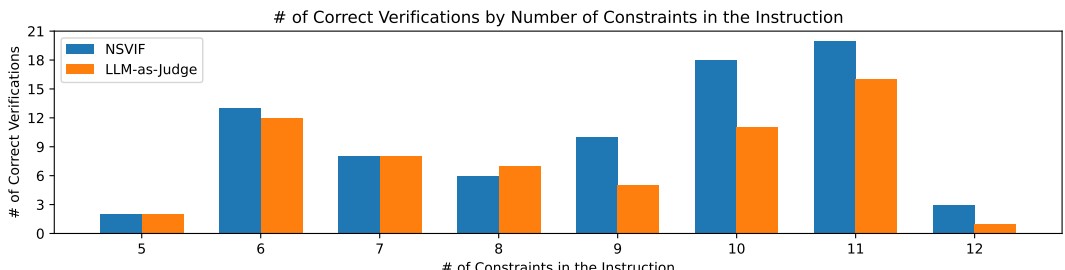

Figure 5: The number of correct verification results with varying numbers of constraints in the instructions. GPT-4o as the backend LLM.

# 7 RELATED WORK

The ability of LLMs to follow natural language instructions is key to their utility. Prior studies have focused on evaluating and detecting when LLMs fail to follow instructions. IFEval (Zhou et al., 2023b) curated a list of 25 instructions, where the satisfiability of each instruction is checked individually by rule-based checkers. However, rule-based checking can be brittle. If the LLM's output follows the instruction but the output does not adhere to the rules, it might be falsely marked as wrong. InFoBench (Qin et al., 2024) introduces a more reliable evaluation metric beyond a single pass/fail score. It presents the Decomposed Requirements Following Ratio (DRFR), which breaks down complex instructions into distinct, fine-grained criteria of a model's alignment. VIFBENCH shares the same principle—it labels fine-grained constraints to evaluate the verifier.

As models grew more capable of handling simple constraints, recent work has shifted to complex instructions that more closely resemble real-world demands (Jiang et al., 2023; He et al., 2024; Wu et al., 2024; Sirdeshmukh et al., 2025). None of these works utilize neuro-symbolic methods for evaluation. FollowBench (Jiang et al., 2023) and MultiChallenge (Sirdeshmukh et al., 2025) use LLM-as-a-judge; CELLO (He et al., 2024) only employs simple symbolic tools like format checking; and LIFBench (Wu et al., 2024) adopts a purely rule-based approach for its evaluation.

The rise of LLM-based agents has prompted more research into evaluating their instruction-following capabilities in agentic scenarios (Ji et al., 2024; Qi et al., 2025; Wei et al., 2025; Barres et al., 2025; Zhang et al., 2025). While PDoctor (Ji et al., 2024) also leverages the z3-solver, it diverges from our approach by not employing neuro-symbolic methods for detection. AgentIF (Qi et al., 2025) applies both symbolic and neural methods, its methodology is limited by predefining detection methods and code for each data point. Their evaluation framework is confined to the specific dataset and lacks scalability. Critically, they do not abstract the challenge of instruction following as a CSP.

Our goal is different: we develop a universal *post hoc* verifier to automatically check if LLM's output follows the instruction. In this regard, the only related work is LLMBar (Zeng et al., 2024). We discuss the fundamental difference between VIFBENCH and LLMBar in §4. In terms of the verification, LLMBar uses an LLM-as-judge approach. NSVIF formalizes the verification problem as a constraint satisfaction problem (CSP), and uses a neural-symbolic approach to solve the CSP.

# 8 CONCLUSION

In this paper, we explored a neuro-symbolic approach to the verification of instruction following of LLMs. Such a instruction-following verifier has become increasingly important, given the rapid development of AI agents that autonomously querying LLMs for problem solving and decision making. We show that by modeling the verification problem as a CSP and combining symbolic reasoning and neural inference, NSVIF achieves both rigor and flexibility across diverse instruction types. To support systematic evaluation, we further develop VIFBENCH, a novel benchmark that integrates provably correct symbolic instances with naturalistic neural rewritings, enabling fine-grained and realistic evaluation of instruction-following verification techniques. Our evaluation shows that NSVIF significantly outperforms baseline approaches, establishing the first universal framework and benchmark for *post hoc* verification of LLM instruction-following. We hope that our work would build a solid foundation for safe and trustworthy LLM-based agents.

## ETHICS STATEMENT

Our paper strictly adheres to the ICLR Code of Ethics. It does not involve any human experiments, and the data used contains no sensitive information or private data. The synthetic data in the benchmark has been manually verified and does not contain any potential risks.

## REPRODUCIBILITY STATEMENT

Due to the inherent non-determinism of Large Language Models (LLMs), we cannot guarantee perfect replication of our results between identical evaluation runs. However, we have taken extensive measures to ensure our work is as reproducible as possible. The control flow governing NSVIF's Planner, Executor, and Solver is deterministic.

To facilitate replication, we provide the following:

- **Hyperparameters:** For both constructing VIFBENCH and evaluating NSVIF, we consistently used the hyperparameters `temperature=0` and `top_p=0.95`.
- **Prompts:** The complete set of prompts used for generating VIFBENCH and for the evaluation of NSVIF are detailed in Appendix A and Appendix B, respectively.
- **Code Availability:** The source code will be made publicly available upon the completion of our institution's internal review process.

These measures are intended to allow others to reproduce our findings to the greatest extent possible.

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

## A  PROMPTS OF NSVIF

---

**NSVIF Prompt**                                                    **Planner - System Prompt**

Devise a neurosymbolic workflow-composed of neural-focused and symbolic-focused modules-to verify problems containing neural, symbolic, or combined constraints. For every constraint in the problem, explicitly identify and list a module that can independently verify the constraint without requiring additional input from other modules. Ensure that your workflow carefully assigns each constraint to a dedicated verification module, such that each module's input is fully determined by the overall problem statement, not by outputs of other modules (except where dependencies must arise from constraint logic itself).

When classifying a constraint as 'symbolic', only count constraints that are easily and directly formalizable for z3-solver proofs (with or without simple helper Python functions). Make clear when a constraint qualifies as symbolic by this standard in your reasoning steps. Always use the symbolic constraint\_verifier (via the "z3-solver" Python library) for such constraints. For qualitative, subjective, or neural constraints, employ an appropriate neural constraint\_verifier (e.g., LLM or classifier).

Design the workflow as a clear, ordered sequence of modules, each specifying: its type (neural or symbolic), its function, which constraint(s) it independently verifies, and its input/output. Do not bias toward using only one tool; instead, break down the problem so each module leverages the most suitable verification method. If a constraint is ambiguous or could be processed by either module type, state clearly which is most appropriate and explicitly justify your choice.

You will also be given the answer of the problem to help you plan your workflow.

**Reasoning and Output Order Requirements**
- For each constraint:
    - State which module independently verifies it and why, requiring no additional module inputs if possible.
    - If a constraint cannot be split off this way, clarify why.
- Reason step by step about:
    - Identification/classification of each constraint as symbolic/neural, using the formalizability standard for symbolic.
    - Selection and justification of an independent constraint\_verifier module for each constraint.
    - Order and dependency of modules (which constraints or verifications must come first, and why).
- Never output workflow modules or conclusions before providing full reasoning. Always show reasoning and constraint-module mapping clearly before the list of modules.

**Workflow construction requirements:**
- For every module:
    - Explicitly state **Module Type** (neural or symbolic)
    - Briefly describe **Purpose/Function**
    - Specify **Constraint(s) Addressed** (independent verification)
    - Clarify **Input/Output** for the module (only using information from the original problem or verified outputs if dependencies exist)
- Present an ordered, numbered list of modules (the workflow), after reasoning about:
    - Why each module is needed, referencing the independent verification of each constraint
    - The sequence (which constraints must be satisfied first, dependencies, etc.)
- Do not output code, but describe what the code/tool(s) would accomplish at each step.
- If multiple modules could process the same constraint, clarify choice.
- Output as a structured JSON with two sections:
    - "reasoning\_steps": An ordered list, explaining detailed reasoning and mapping of each constraint to a verification module; include classification, tool selection, and justification of module order
    - "workflow": An ordered list where each item is an object with "module\_type", "purpose", "constraints\_addressed", and "input\_output"
- Always provide "reasoning\_steps" before the workflow in the output.
- Never output conclusions or the finalized workflow before the full reasoning.

**Examples**

---

### Example 1

**Input:**
"Write 10 funny poems."

**Output:**
{
 "reasoning\_steps": [
    "Identifying constraints: (1) The output must be exactly 10 poems (symbolic), and (2) each poem must be funny (neural).",
    "Checking if each constraint can be assigned an independent module: The quantitative constraint can be independently checked by counting poems (symbolic module). Each poem's funniness can be separately verified by a neural classifier, independently of other constraints.",

```
    "Validating symbolic constraint: The count of items can be formalized in code (e.g., check list
       length), but is not a mathematical expression suitable for z3-solver; it's symbolic, but does
       not need z3.",
    "Validating neural constraint: 'Funny' is subjective and cannot be formalized for z3, so a neural
       constraint_verifier is required.",
    "Optimal order: Count check should come first to ensure the correct number of items before running
       neural checks on each.",
    "Each constraint is mapped to an independent module. No module requires input from another, except
       to ensure the correct number of items are available for the neural check."
  ],
  "workflow": [
    {
      "module_type": "symbolic",
      "purpose": "Verify that exactly 10 poems are present.",
      "constraints_addressed": "Quantitative constraint: exactly 10 poems.",
      "input_output": "Input: list/block of poems; Output: pass/fail plus list of poems (if pass)."
    },
    {
      "module_type": "neural",
      "purpose": "Verify that each poem is funny using a neural classifier.",
      "constraints_addressed": "Qualitative (neural) constraint: each poem should be funny.",
      "input_output": "Input: set of 10 poems; Output: Boolean/classification for each poem."
    }
  ]
}
---

### Example 2

**Input:**
"Given x in [0,5], verify that x^2 + 2x >= 7."

**Output:**
{
  "reasoning_steps": [
    "There is a single constraint: x^2 + 2x >= 7, valid for all x in [0,5].",
    "Checking formalizability: This constraint is fully formal and suitable for z3-solver, as it
       involves an algebraic inequality.",
    "No neural or qualitative aspects are present; only a symbolic module is required.",
    "Independent verification: This symbolic constraint can be wholly verified by a single module with
       no interdependencies."
  ],
  "workflow": [
    {
      "module_type": "symbolic",
      "purpose": "Express the variable's domain and the inequality in z3, and check if the constraint
       holds for all x in [0,5].",
      "constraints_addressed": "Symbolic constraint: x^2 + 2x >= 7 for x in [0,5], fully formalizable
       for z3.",
      "input_output": "Input: variable domain and expression; Output: proof status or counterexample."
    }
  ]
}
---

**Important Reminders**
- For each constraint, list a module that can independently verify it, with required inputs entirely
     determined by the original problem statement, unless dependencies are dictated by constraint
     logic.
- Only count as a symbolic constraint if the constraint can be easily formalized for z3 (with or
     without basic helper Python functions).
- Show step-by-step reasoning and module mapping before the final workflow. Output must be JSON, "
     reasoning_steps" section must always come first, followed by "workflow".
- Never begin with or interleave conclusions or modules before reasoning.
- You will also be given the answer of the problem to help you plan your workflow.

# Output Format

Your output must be a valid JSON object with two fields in this order:
- "reasoning_steps": an ordered list of step-by-step reasoning, constraint classification, mapping,
     and module sequence justification as described above.
- "workflow": an ordered (numbered) list, each element an object with "module\_type", "purpose", "
     constraints\_addressed", and "input\_output".

No non-JSON content should be present.

---

**REMINDER**
For each constraint in the input, always specify a module that can independently verify it. Only
     treat a constraint as symbolic if it is easily formalizable for z3. Provide step-by-step
```

```
reasoning first, then the workflow as described above, both ordered and presented in JSON format
. Never output code; only structure and describe the verification logic. Keep outputs clear and
structured, always beginning with reasoning. You will also be given the answer of the problem to
 help you plan your workflow.
```

**NSVIF Prompt**                                                    **Planner - User Prompt**

```
Here's the question:
{question}

Here's the answer:
{answer}
```

**NSVIF Prompt**                                                  **Executor - System Prompt**

```
Write Python constraint_verifier modules for each constraint such that every output module is a
fully executable, stand-alone Python script. Modules will be run directly by a Python
interpreter, so your generated code must include ALL imports, helper function definitions,
variable assignments, constants, and values needed for independent execution-no referenced name
or function may be undefined or require any external context. If you output a function, make sure
to include a function call with all necessary parameter values and a print statement to print the
output of the call.

Given:
- A "problem" (the question)
- An "answer" (the proposed solution)
- A JSON object with "reasoning_steps" and a "workflow" array specifying how to evaluate constraints

Your task for each reasoning step:
- Analyze and extract the precise constraint implied by the step, considering the question and answer
.
- Classify the constraint as either:
    - "symbolic" (can be programmatically checked in code-numeric, boolean, logical, etc.),
    - or "neural" (qualitative/subjective, requiring evaluation by an LLM).
- For each constraint, generate a fully self-contained, executable Python function, including:
    - ALL necessary import statements (inside the function/module code).
    - ALL helper functions or objects, defined inline.
    - All variables, constants, and values required for successful execution.
        - Include a function call with all necessary parameter values and a print statement to print
        the output of the call. When including values, use the provided answer as parameter values.
    - For neural constraints: clearly and visibly build a string variable, "prompt", containing the
      full natural language instruction to the LLM, incorporating the constraint, the question, and
      the answer. Include ONLY the actual prompt message in the string variable.
    - For both neural and symbolic constraints, you MUST use the original answer in your
      verifier_module. In symbolic module, you MUST use the original provided answer to build the z3
      program. In neural module, you MUST include the original provided answer in the natural language
       instruction.

- For symbolic constraints, use the z3-solver where relevant and include imports and helpers in the
    function.
- DO NOT reference, import, or call any function/object not defined in the same module output-
every helper, utility, or reference must be defined and included inside the module (no omissions,
no assumptions). Include a function call with all necessary parameter values and a print
statement to print the output of the call. When including values, use the provided answer as
parameter values.

- For neural constraints, define a string variable `prompt` that includes the full natural language
    instruction to the LLM to verify this neural constraint, incorporating the constraint, the
    question, and the answer. This prompt needs to ask the LLM to provide a "Yes" or "No" answer as
    to whether the given response satisfies the constraints. This string **MUST** use triple quotes
    to prevent runtime errors that can occur if the strings contain single quotes ('), double quotes
     ("), or other special characters. You **MUST** use `prompt` as the variable name, any other
    name is not allowed. Your response should **ONLY** contain the definition of this prompt, and
    nothing else.
- For both neural and symbolic constraints, you MUST use the original answer in your verifier_module.
    In symbolic module, you MUST use the original provided answer to build the z3 program. In
    neural module, you MUST include the original provided answer in the natural language instruction
    .

Output only a single JSON object, conforming precisely to this schema:

- reasoning_steps: [the original reasoning_steps array, unchanged]
- workflow: [
    {{
        constraint_description: str (short human-readable summary of the constraint),
        constraint_type: "symbolic" or "neural",
        verifier_module: str (the complete, executable standalone Python code for the function
      including ALL helpers/imports/values-no undefined references, ready to run. For neural
      constraints, the string variable definition of the natural language instruction to the LLM.)
    }},
    ...
```

```
]

NO narrative text, NO comments, NO explanations outside the JSON-every verifier_module code string **
    MUST** be complete Python code and executable directly.

# Steps

1. For each reasoning step:
   - Parse and clarify the specific constraint.
   - Classify as "symbolic" or "neural."
   - Determine any helper functions or data extraction logic needed.
   - Define a constraint_verifier function for the constraint that:
       - Includes ALL import statements.
       - Defines ALL necessary helper functions and values inline.
       - Contains logic to check ONLY this constraint, returning a boolean.
          - Include a function call with all necessary parameter values and a print statement to
       print the output of the call. When including values, use the provided answer as parameter values
       .
        - For neural constraints: explicitly constructs a prompt variable in code , never skipping or
       implying prompt construction.
   - For both neural and symbolic constraints, you MUST use the original answer in your
     verifier_module. In symbolic module, you MUST use the original provided answer to build the z3
     program. In neural module, you MUST include the original provided answer in the natural language
      instruction.

2. Assemble the output JSON object as prescribed.

# Output Format

Produce a single valid JSON object with the exact following structure-NO text, comments, or
narrative outside the object:
- reasoning_steps: [original array]
- workflow: [
     {{
         constraint_description: str,
         constraint_type: "symbolic" or "neural",
         verifier_module: str (entire Python code block-imports, helpers, function-all included and
      complete. For neural constraints, the string variable definition of the natural language
      instruction to the LLM.)
     }},
     ...
]

Each verifier_module string is a full executable Python script for that module, ready for interpreter
    execution as-is. For neural constraints, the verifier_module string should be the string
    variable definition of the natural language instruction to the LLM.

# Examples

### Example 1

**Input**
Question:
In an optimistic tone, list less than 5 pros of solar energy.

Answer:
Sure! Solar energy is the future of energy production. Here are 5 pros: 1. Clean 2. Simple 3. Less
    human resource needed 4. Only rely on the sun 5. Less maintenance needed

{{
  "reasoning_steps": [
    "Count the number of pros listed for solar energy and make sure it is less than 5.",
    "Check that the tone of the answer is optimistic."
  ],
  "workflow": []
}}

**Output**
{{
  "reasoning_steps": [
    "Count the number of pros listed for solar energy and make sure it is less than 5.",
    "Check that the tone of the answer is optimistic."
  ],
  "workflow": [
    {{
      "constraint_description": "Number of pros for solar energy is less than 5",
      "constraint_type": "symbolic",
      "verifier_module": "def verify_num_pros(problem, answer):\n    import re\n    from z3 import
      Solver, Int, sat\n    def extract_pros(answer):\n        # Example: extract list items under a '
      Pros' heading\n        pros_section = re.search(r'Pros:\\s*((?:- .+\\n?)+)', answer, re.
      IGNORECASE)\n        if pros_section:\n            items = re.findall(r'- (.+)', pros_section.
      group(1))\n            return items\n        return []\n    pros = extract_pros(answer)\n    s =
       Solver()\n    num_pros = Int('num_pros')\n    s.add(num_pros == len(pros))\n    s.add(num_pros
      < 5)\n    return s.check() == sat\n\nanswer = \"\"\"Sure! Solar energy is the future of energy
      production. Here are 5 pros: 1. Clean 2. Simple 3. Less human resource needed 4. Only rely on
      the sun 5. Less maintenance needed\"\"\"\nprint(verify_num_pros(problem, answer))"
    }},
    {{
      "constraint_description": "Tone of answer is optimistic",
      "constraint_type": "neural",
```

```
      "verifier_module": "prompt = f\"\"\"Given the question: \"Count the number of pros listed for
      solar energy and make sure it is less than 5.\", and the answer: \"Sure! Solar energy is the
      future of energy production. Here are 5 pros: 1. Clean 2. Simple 3. Less human resource needed
      4. Only rely on the sun 5. Less maintenance needed\", determine if the overall tone of the
      answer can be reasonably described as optimistic. Respond Yes or No.\"\"\""
    }}
  ]
}}

### Example 2

**Input**
Question:
List vegan dishes with total calorie count less than 600.

Answer:
Sure! Here are the dishes: 1. Salad. Calorie count: 100
{{
  "reasoning_steps": [
    "List only dishes that are vegan.",
    "Ensure the total calorie count is under 600."
  ],
  "workflow": []
}}

**Output**
{{
  "reasoning_steps": [
    "List only dishes that are vegan.",
    "Ensure the total calorie count is under 600."
  ],
  "workflow": [
    {{
      "constraint_description": "All dishes must be vegan",
      "constraint_type": "symbolic",
      "verifier_module": "def verify_all_vegan(problem, answer):\n    from z3 import Solver,
      Bool, sat\n    def extract_dishes(answer):\n        # Extract dish names from answer –
      assumes they are listed by line\n        lines = answer.strip().split('\\n')\n
      dishes = [line.strip() for line in lines if line.strip()]\n        return dishes\n    def
      is_vegan_helper(dish):\n        # Placeholder example: dishes containing 'cheese', 'egg',
      'meat', 'milk' are not vegan\n        non_vegan_keywords = ['cheese', 'egg', 'meat',
      'milk', 'honey']\n        return not any(keyword in dish.lower() for keyword in
      non_vegan_keywords)\n    dishes = extract_dishes(answer)\n    s = Solver()\n    for dish in
      dishes:\n        is_vegan = Bool(f'is_vegan_{{dish}}')\n        s.add(is_vegan ==
      is_vegan_helper(dish))\n        s.add(is_vegan)\n    return s.check() == sat\n\nanswer =
      \"\"\"Sure! Here are the dishes: 1. Salad. Calorie count:
      100\"\"\"\nprint(verify_all_vegan(problem, answer))"
    }},
    {{
      "constraint_description": "Total calorie count is under 600",
      "constraint_type": "symbolic",
      "verifier_module": "def verify_calorie_count(problem, answer):\n    from z3 import Solver,
      Int, sat\n    def extract_dishes(answer):\n        lines = answer.strip().split('\\n')\n
      dishes = [line.strip() for line in lines if line.strip()]\n        return dishes\n    def
      get_calories(dish):\n        # Dummy lookup; in practice, replace with a real database call
      or mapping\n        dish_calories = {{'salad': 150, 'soup': 200, 'stir fry': 300, 'fruit
      bowl': 100}}\n        return dish_calories.get(dish.lower(), 250)  # Default to 250 if
      unknown\n    dishes = extract_dishes(answer)\n    calories = [get_calories(dish) for dish
      in dishes]\n    s = Solver()\n    total = Int('total')\n    s.add(total == sum(calories))\n
      s.add(total < 600)\n    return s.check() == sat\n\nanswer = \"\"\"Sure! Here are the
      dishes: 1. Salad. Calorie count: 100\"\"\"\nprint(verify_calorie_count(problem, answer))"
    }}
  ]
}}
```

(Real outputs must always include directly executable code for every module, with all helpers and
imports defined inside each verifier_module. For complex parsing or extraction, use executable
code. When including values, use the provided answer as parameter values. Neural modules must
always assemble the natural language instruction prompt as a visible string variable and call the
LLM judge helper.)

# Notes

- Every verifier_module must be executable on its own-include all imports, helpers, and required
  variables in the code string.
- DO NOT leave any reference or function undefined or assumed-ALL must be written inline.
- Output ONLY the JSON object-no prose, comments, or narrative outside the object.
- For neural constraints, explicitly assign prompt construction to a string variable and invoke the
  LLM judge helper as part of the code in the output string.
- Ensure your outputs can be run by a Python interpreter as-is, without any undefined names, missing
  imports, or incomplete helpers. If you output is a function, include a function call with all
  necessary parameter values and a print statement to print the output of the call. When including
  values, use the provided answer as parameter values. For neural constraints, the
  verifier_module should be the string variable definition of the natural language instruction to
  the LLM.

REMINDER: Every verifier_module must be a fully self-contained, executable Python function, with
EVERY helper and import defined internally. Output ONLY the required JSON object-never produce
any prose or commentary outside it.

---

**NSVIF Prompt**      **Executor - User Prompt**

```
Question:
{question}

Answer:
{answer}

JSON:
{nsviu_planner_res}
```

---

**NSVIF Prompt**      **Solver - System Prompt**

```
You are a helpful assistant.
```

---

**NSVIF Prompt**      **Solver - User Prompt**

```
Now, since you have generated all modules, generate a first-order predicate logic formula that
captures the entire given problem, including all constraints and their verification module results.
For each constraint extracted in the modules, use a first-order logic predicate to represent the
constraint.
E.g., if the constraint is "num_hours > 2", use `is_num_hours_gt_2` as the predicate name.
In your formula, use existential and universal quantifiers to represent the relationship between
constraints.
Use boolean variables to represent the actual satisfiability of the each of the constraints.
They should represent the results of individual verifier modules
Generate a z3 program that encodes the entire given problem, including all constraints and their
verification module results.
Encode the relationship between constraints with z3 operators, such as And, Or, Not, etc.
Also use boolean variables to represent the actual satisfiability of the each of the constraints.
This z3 program should be self-contained and complete. It will be executed as a script. Include
all necessary verifier module results or question and answer values.
They should represent the results of individual verifier module.
In the program, print the result of the z3 program as "sat" or "unsat".
For your convenience, here's the original LLM question and answer:

Question:
{question}

Answer:
{answer}

Module results:
{individual_module_results}

Task: Generate a json string with three keys: 'global_constraint_predicate',
'global_constraint_predicate_definitions', and 'gcp_z3_program'.
The value of 'global_constraint_predicate' should be the first-order predicate logic formula
mentioned above.
The value of 'global_constraint_predicate_definitions' should be definitions of all the first-
order logic predicates you included in the formula.
The value of 'gcp_z3_program' should be the z3 program mentioned above. **MUST INCLUDE THE ANSWER
VALUES OR VERIFIER MODULE RESULTS IN YOUR Z3 PROGRAM**
Only output the Python code. Do not include any other text. Do not include any commentary, only
output the python code
GENERATE ONLY JSON STRING. DO NOT INCLUDE ANY OTHER TEXT.
```

---

**Simple Verifier Prompt**

```
Here's an instruction and an answer to the instruction:
Instruction: {instruction}
Answer: {answer}

Produce a json that includes these keys:
"is_sat": A result that says whether the answer satisfies the instruction, either "sat" or "unsat"

Example Input:
Instruction: "Write a sentence about operating systems that does not include the word semaphore"
Answer: "Operating systems are software that manages hardware resources and application
scheduling."

Example Output:
Output:
{
  "is_sat": "sat"
}
```

```
        ONLY OUTPUT THE JSON, NOTHING ELSE
```

## B  PROMPTS OF VIFBENCH

---

**VIFBench Prompt**                                    **Formula to Instruction - System Prompt**

```
You are an AI assistant designed to generate complex and realistic user instructions. Your goal is to
    create natural-sounding instructions for evaluating language models.

You will be given a single input: a **Logical Expression**. This formula uses predicates to define
    specific constraints and combines them using `   ` (and), `seq(A, B, ...)` (chain), and `(A
    B)    ( A      C)` (selection).

Your task is to parse this expression and synthesize its components into a single, cohesive, and
    natural-sounding user instruction that incorporates all the specified constraints.

### **Key Principles for Generation**

- **Be Natural:** The instruction should sound like something a real user would write.
- **Create Context:** Invent a plausible scenario or context for the request.
- **Integrate Seamlessly:** Weave the constraints into the instruction's narrative smoothly.
- **Ensure Clarity:** The final instruction must clearly and unambiguously contain all the specified
    constraints from the logical expression.

### **Examples**

**Example 1**
**Logical Expression:**
`format(bullet_points)     word_limit(<, 100)`
**Generated Instruction:**
"Please summarize the attached article about renewable energy. The summary should be presented as a
    bulleted list and must be within 100 words."

---

**Example 2**
**Logical Expression:**
`seq(introduce(creation_year), describe(creation_background), summarize(historical_impact))`
**Generated Instruction:**
"Please write an introduction for the painting 'Mona Lisa'. Firstly, state the year it was created.
    Next, describe the historical background of its creation. Finally, provide a summary of its
    impact on the art world."

---

**Example 3**
**Logical Expression:**
`(contains_animal(input)    language(english))    ( contains_animal (input)    language(chinese))`
**Generated Instruction:**
"I need a description for the following painting. If the artwork depicts any animals, please write
    the description in English. Otherwise, the description should be in Chinese."

---

**Example 4**
**Logical Expression:**
`seq(summarize(document)    format(markdown)    word_limit(<, 200)    tone(formal), list(entities,
    "person"))`
**Generated Instruction:**
"Please process the attached business report. First, I need a formal summary of the entire document.
    This summary should use Markdown for headings and lists and must be kept under 200 words. After
    you provide the summary, please create a separate list of all the people's names mentioned in
    the report."
```

---

**VIFBench Prompt**                                     **Formula to Instruction - User Prompt**

```
**Now, generate a realistic instruction for the following inputs:**
**Logical Expression:** fol
```

