# OpenReview forum: "Post Hoc Neuro-Symbolic Verification on Instruction Following of Language Models"
_ICLR.cc/2026/Conference — ICLR 2026 Conference Withdrawn Submission_

### Official Review · Reviewer_dBEo · 2025-11-01

**Soundness:** 2
**Presentation:** 3
**Contribution:** 2
**Rating:** 4
**Confidence:** 3

**Summary:**

This paper present a neurosymbolic verifier to check whether an LLM-generated response satisfies the constraints indicated in the input. It does so by generating a set of formal and neural properties to check for, and then checks if the generation satisfies the properties.

**Strengths:**

- The overall idea is clear (though there are questions about details, see below)
- The benchmark seems cool

**Weaknesses:**

- It is unclear how a neural verifier can truly be used for "verification" - at best it might be predictive, and with sufficient empirical testing for calibration, could provide some form of statistical results, but I wouldn't call it verification.
- The "Formula Contextualization" presentation is too vague to understand what is going on. It is unclear how the first-order predicates are grounded, or evaluated over the inputs.
- The compositional verification formula is not quite sound, or not accurately presented: it should be checking for all the constraints, but the fourth equation (unnumbered) just seems to check one?
- The novelty of the approach is not clear wrt the following papers. A better discussion of novelty is in order.

Olausson, Theo, Alex Gu, Ben Lipkin, Cedegao Zhang, Armando Solar-Lezama, Joshua Tenenbaum, and Roger Levy. "LINC: A Neurosymbolic Approach for Logical Reasoning by Combining Language Models with First-Order Logic Provers." In Proceedings of the 2023 Conference on Empirical Methods in Natural Language Processing, pp. 5153-5176. 2023.

Zhang, Yedi, Yufan Cai, Xinyue Zuo, Xiaokun Luan, Kailong Wang, Zhe Hou, Yifan Zhang et al. "Position: Trustworthy AI Agents Require the Integration of Large Language Models and Formal Methods." In Forty-second International Conference on Machine Learning Position Paper Track.

**Questions:**

- How are the first order literals in the formulas grounded?
- How is a neural verifier actually guaranteed?
- What is the novelty of this paper wrt previous work?

---

> ### Author Response · Authors · 2025-11-21
>
> Thank you for your detailed and comprehensive feedback. We answer your questions below:
>
> > W1: It is unclear how a neural verifier can truly be used for "verification" - at best it might be predictive, and with sufficient empirical testing for calibration, could provide some form of statistical results, but I wouldn't call it verification.
>
> We appreciate the opportunity to clarify the meaning of "verification" in our work. We fully agree that the term 'verification' carries strong connotations from the formal methods domain. To be precise, our use of the term here does not refer to formal verification (i.e., deriving guaranteed or provable outcomes through rigorous deduction or proof). Instead, 'verification' is used in the context of empirical evaluation and compliance checking. It refers specifically to our goal of validating whether the LLM's generated response complies with the specific constraints in the given instruction.
>
> > Q1: How are the first order literals in the formulas grounded?
>
> We can clarify this using Figure 2 as an illustrative example. In the formula, the variable $x$ is grounded to the entire generated LLM response text. Therefore, all predicates operate on the complete output. The final verification formula is constructed as a conjunction of multiple predicates. Each of these predicates is grounded by mapping it to its dedicated verification module.
>
> > Q2: How is a neural verifier actually guaranteed?
>
> We want to be explicit: our neural verifier does not provide a formal guarantee or certifiably accurate results in the sense of formal verification. However, we argue that its reliability is significantly superior to direct 'LLM-as-a-Judge' methodology. By modeling instruction following verification as a Constraint Satisfaction Problem (CSP), our verifier analyzes the satisfaction of each individual constraint separately, ensuring accurate and interpretable evaluation.
>
> > Q3: What is the novelty of this paper wrt previous work?
>
> While we acknowledge that LINC also utilizes a neuro-symbolic approach, its fundamental objective is distinct from ours. Specifically, LINC is focused on enhancing the logical reasoning capability of LLMs. In contrast, our paper addresses the instruction following verification problem: we introduce the first framework designed to robustly and formally verify the compliance of LLM outputs against complex instruction. Our contribution lies in the evaluation methodology, not the model generation itself.
>
> Regarding Position, as it is a review article, its scope is primarily descriptive. It does not propose a novel verification framework or a benchmark directly addressing the challenge of instruction following verification, making its direct relevance to our technical contribution limited.

---

> > ### Comment · Reviewer_dBEo · 2025-11-21
> > **Resolving Clarification**
> >
> > Thank you for the responses.
> >
> > I'd call the result something like "empirical verification" or "validation" rather than "verification", since the latter implies strong connotations as you rightly pointed out.
> >
> > The relation to the two papers sounds good. The comparison to LINC should be included in the paper - and more broadly making the point as in the response that while there exist previous neurosymbolic approaches (with citations) to augment LLM reasoning, the objective of this work is to perform empirical verification.
> >
> > Q1: I am still unclear on how the grounding works. The paper does not clearly explain 1) how the literals are enumerated, and 2) how the mapping from the literals to the generation is grounded. For example, in figure 1b, the formula is `num_pros=3`. How is the literal `num_pros` grounded? How is the list of literals (`num_pros`, `num_cons`), etc. enumerated? It would be good to explain this procedure precisely.

---

### Official Review · Reviewer_Kda5 · 2025-11-01

**Soundness:** 3
**Presentation:** 3
**Contribution:** 2
**Rating:** 2
**Confidence:** 4

**Summary:**

The papers aims to address the issue of verifying whether LLMs follow instructions given through post-hoc verification. It presents a high-level idea of encoding the given input text as a set of logical constraints, some symbolic and some neural, which are then jointly checked for on the output resulting in a system they call NSVIF. The work creates a new benchmark called VIFBENCH which contains logical specifications for the given tasks. This benchmark is then used to test the hypothesis that NSVIF is better at verifying the derived constraints than the baseline approach of using 'LLM-as-a-judge'. The results show that NSVIF does better than the baseline by about 20-25% in precision and 40% in recall.

**Strengths:**

- VIFBENCH is a nice concrete outcome of the paper, which could be used in evaluation in other works.

- The general plan outlined in Section 3 and Figure 2 aims for a modular pipeline. The purely symbolic constraints are checked with the Z3 SMT solver whereas the subjective constraints using LLMs.

- The overall direction on verifying outputs of LLMs using symbolic methods, at least partially, is good.

- The results are positive, although the experiments run are partial and don't test the hypothesis fully. The paper explains the suspected reasons why the evaluated LLMs get stuck and possibly why they don't perform well.

**Weaknesses:**

- The main text of the paper does not give much details about VIFBENCH. It says the benchmark is comprehensive but provides little justification for why so. One of the challenges in such evaluations is to avoid biasing the benchmarks to a particular outcome. How did you avoid it?

- The most problematic part of the paper is Section 3. Beyond giving a high-level architectural roadmap, there are almost no details on how and why NSVIF is the way it is. This makes it difficult to draw well-grounded conclusions from the reported experiments in the paper. I will highlight several sources of unsoundness next:

1) There are vague statements "use a neural parser to analyze and parse the implicit constraints" that elide details. What is implicit constraints? How did you check that these constraints are accurately derived for the benchmarks or test subjects?

2) A common issue when trying to solve neuro-symbolic constraints is the deal with variables that are constrained in both symbolic and subjective constraints (see for example early works like [a] which highlight the challenge in Section 3 and 4). How are the constraints really encoded? It is possible to encode them either as symbolic or as subjective ones. How do you choose? Multiple encodings in either to be used but the paper is quite thin on technical details.

3) Another significant source of errors in your design is with the executor. The executor relies on LLMs and best-effort loop to check the constraint satisfaction, but its incomplete and the paper doesn't quantify how this affects the final conclusions drawn.

4) Another issue is in formula contextualization. The neural paraphraser can get things wrong or over-concretize things to a specific context. This appears to be a somewhat arbitrary choice in the methodology. When do you decide between concretizing values for a symbolic variable vs. keeping it symbolic? E.g. for "sports", does the paraphrases specialize it to a certain sport keep it fully abstract? The difference matters because providing the first-order logic formula might change the outcome significantly.

- The evaluation methodology is incomplete and quite problematic in its rigor in eliminating sources of unsoundness.

1) Models that infinitely think or fail to respond cause issues in the approach. But they could have been eliminated or substituted with outputs of other models.

2) The analysis of the experiments states that "LLM-based verifiers do not understand constraint instructions at all." I am not sure whether this statement is true of all such verifiers. And if so, then what is the point of reporting on the final result which is clearly conditioned on this working correctly but requires a lot else to work.

3)  LLMs are reported as inconsistent due to non-determinism. The evaluation results present no standard deviations to quantify the same.

I understand that the experimental methodology has limitation inherent of evaluating black-box LLMs. Nonetheless, it is also difficult to draw scientific conclusions when the experimental methodology ran into so many internal inconsistencies.



[a] "Neuro-Symbolic Execution: Augmenting Symbolic Execution with Neural Constraints", Shen et al., In NDSS 2019.

**Questions:**

- Can you clarify why you believe the conclusion you draw is accurate, despite the numerous ways in which evaluation methodology itself could fail? You have stated several ways upfront in the paper in how so.

- LLMs are used in NSVIF such as for a neural parser. Is this the same as that evaluated against in 'LLM-as-a-judge' baseline? What issues do you foresee in using the same LLM in both vs. different ones?

---

> ### Author Response · Authors · 2025-11-21
>
> Thank you for your constructive and insightful comments. We respond to each of your questions below:
>
> > W1: It says the benchmark is comprehensive but provides little justification for why so.
>
> We appreciate your keen observation regarding the lack of detailed justification for the benchmark's comprehensiveness. We consider our benchmark comprehensive because the instruction samples are curated to exhaustively cover all instruction-following failure phenomena described in Table 1.
>
> > W2: One of the challenges in such evaluations is to avoid biasing the benchmarks to a particular outcome. How did you avoid it?
>
> We agree that preventing outcome bias is a critical challenge in such evaluations, and we addressed this by ensuring a nearly balanced distribution of outcomes. Each instruction in our benchmark includes a corresponding response, paired with a human-verified sat/unsat label. To ensure the final outcome distribution did not bias towards sat or unsat, we selectively re-generated responses belonging to the majority class until the counts of sat and unsat instances were approximately equal. The final sat/unsat ratio in our dataset is 1.14, confirming a highly controlled and balanced outcome distribution.
>
> > W3: There are vague statements "use a neural parser to analyze and parse the implicit constraints" that elide details. What is implicit constraints? How did you check that these constraints are accurately derived for the benchmarks or test subjects?
>
> The term 'implicit constraints' refers to constraints that are implied or encoded within the natural language instruction, but are not clearly listed as direct commands.
>
> > W4: How are the constraints really encoded? It is possible to encode them either as symbolic or as subjective ones. How do you choose? Multiple encodings in either to be used but the paper is quite thin on technical details.
>
> We recognize that the paper was thin on these technical details and appreciate the opportunity to clarify our encoding strategy.
>
> For VIFBENCH (our benchmark dataset), all constraints are meticulously designed to be mutually exclusive. Each constraint is unambiguously classified and encoded as either purely symbolic or purely neural.
>
> In the context of the NSVIF framework's application to general instructions—where a constraint could be interpreted as having a neuro-symbolic nature—our policy is to encode it exclusively as either symbolic or neural, based on our neural parser's choice. This decision is governed by a clear prioritization rule: our neural parser is designed to identify and encode a constraint as symbolic whenever possible (e.g., length, format), only defaulting to neural encoding when the constraint inherently requires semantic understanding or subjective evaluation.
>
> > W5: When do you decide between concretizing values for a symbolic variable vs. keeping it symbolic? E.g. for "sports", does the paraphrases specialize it to a certain sport keep it fully abstract? The difference matters because providing the first-order logic formula might change the outcome significantly.
>
> We confirm that the variability is inherent to our construction process: since the paraphrasing is performed by the LLM, the model may contextualize an abstract concept (like 'sports') into a specific instance (e.g., 'basketball') or retain the abstract category.
>
> However, we argue that this variability is not detrimental to the construction of our benchmark; instead, it is a desirable feature. By including both general and specialized constraints, our benchmark can test:
>
> - Abstract constraints: instructions containing a general constraint like 'find some materials'
>
> - Concrete constraints: instructions containing a specific constraint like 'find some papers on arXiv'

---

### Official Review · Reviewer_4SfN · 2025-11-01

**Soundness:** 3
**Presentation:** 3
**Contribution:** 3
**Rating:** 2
**Confidence:** 4

**Summary:**

This paper proposes a technique and benchmark for verifying instruction following of LLMs. They propose a technique that breaks the constraints into two kinds: symbolic constraints, and neural constraints, each of which are verified by independent modules that then get combined and verified together via an SMT program. They also propose a new benchmark that contains verifiable constraints to accurately evaluate how well their technique can verify the output of an LLM. They conclude with an evaluation of their technique in comparison with an LLM as a judge.

**Strengths:**

1. The authors are addressing an important problem with LLM evaluations, which is how do we know that all the instructions from the prompt are being followed. I also appreciate the construction of a benchmark that specializes in doing so.
2. The technique seems quite sound, with the authors offloading most of the reasoning parts to the SMT solver, while using LLMs to extract and validate intermediate constraints.

**Weaknesses:**

1. Evaluation: I feel that the evaluation baselines are a bit lacking. First off, not enough models are used here in the evaluations. There are only 2 families of models being evaluated, non of which are open-source, and only one family of models is being used with NSVIF. I would like to see a more thorough evaluation. Second, I find it very hard to believe that *all* 4 models simply declare the outputs as satisfying the instructions, and on closer observation, I see that the prompt is a one-shot prompt with only 'sat' as an output in the examples. This seems suspicious: at best, this baseline is a strawman that is instructed to only output sat unless you actually give an example of what 'unsat' means in the prompt. I am fine with the prompt being two-shot to achieve this. Third, you can come up with another baseline that takes an autoformalization approach: give the LLM the instruction and the answer, and ask it to come up with an SMT program to verify the answer. If the LLM cannot do so, that further justifies a need for NSVIF, but I would like to see that experiment.

2. Some details are missing from the benchmark creation section of the paper. How do you come up with the initial seed predicates? What is the distribution of neural and symbolic predicates? Did you try disjuncting constraints? What is the final distribution of the generated problems with respect to constraint complexity? How do you come up with symbolic predicates v/s neural predicates (e.g. do you actually write the symbolic module that validates the predicate)?

**Questions:**

See weaknesses. Also:
1. Did you try if LLMs can show more improvements on tasks if given feedback via NSVIF v/s LLM as a judge?

---

> ### Author Response · Authors · 2025-11-21
>
> Thank you for your dedicated time and effort in thoroughly reviewing our work. Due to the current constraints of this review cycle, we are unable to integrate these comprehensive new experiments at this time. However, we are committed to thoroughly incorporating these valuable suggestions. We will conduct the requested experiments and include a detailed analysis in the next revised version of this work. We respond to some of your questions below:
>
> > W1: How do you come up with the initial seed predicates?
>
> We randomly sample categories from Table 1 and use an LLM to convert them into specific predicates. For example, a sampled 'Structural Violation' is converted into a predicate like `BulletPoint(x)`.
>
> > W2: What is the distribution of neural and symbolic predicates?
>
> Our dataset maintains a relatively balanced distribution between the two types. On average, each instruction contains 9.91 predicates, consisting of 5.34 neural and 4.56 symbolic predicates (a symbolic-to-neural ratio of 0.85).
>
> > W3: Did you try disjuncting constraints?
>
> Yes. Here is an example instruction containing such constraints disjunction from our benchmark: `If your response is around 120 words (within 10 words difference is ok), please write it from the perspective of a project manager; otherwise, write in a comedian' perspective and technical style.`
>
> > W4: What is the final distribution of the generated problems with respect to constraint complexity?
>
> The distribution is tightly clustered around the mean of 9.91, indicated by a low standard deviation of 1.13. The number of predicates per instruction range from a minimum of 7 to a maximum of 12.
>
> > W5: Do you actually write the symbolic module that validates the predicate?
>
> We did not implement an automated symbolic validation module. The correctness of the predicates generated by the LLM was ensured through manual human inspection.

---

> > ### Comment · Reviewer_4SfN · 2025-11-27
> >
> > Thanks for addressing my issues. I'll maintain my score for now, since I feel that the thorough evaluation is vital for the credibility of the technique. Some additional questions:
> >
> > 1. Did you figure out the reason why all the models simply declared the outputs as sat? Was it indeed the prompt construction?

---

### Note · Authors · 2025-12-31

**Comment:**

The current version is not yet ready and will be substantially revised before resubmission. Thank you for your time.

**Withdrawal Confirmation:**

I have read and agree with the venue's withdrawal policy on behalf of myself and my co-authors.